# Identifying optimal COVID-19 testing strategies for schools and businesses: Balancing testing frequency, individual test technology, and cost

Gregory D. Lyng[1], Natalie E. Sheils[1], Caleb J. Kennedy[1], Daniel O. Griffin[2,3], Ethan M. Berke[1] *

1 OptumLabs, UnitedHealth Group, Minnetonka, MN, United States of America, 2 Division of Infectious Diseases, Department of Medicine, Columbia University, New York, NY, United States of America, 3 ProHealth Care, Optum, Lake Success, NY, United States of America

* ethan.berke@uhg.com

**Data Availability Statement:** Mathematical code is available in the supplemental file submitted with the manuscript.

## Abstract

### Background

COVID-19 test sensitivity and specificity have been widely examined and discussed, yet optimal use of these tests will depend on the goals of testing, the population or setting, and the anticipated underlying disease prevalence. We model various combinations of key variables to identify and compare a range of effective and practical surveillance strategies for schools and businesses.

### Methods

We coupled a simulated data set incorporating actual community prevalence and test performance characteristics to a susceptible, infectious, removed (SIR) compartmental model, modeling the impact of base and tunable variables including test sensitivity, testing frequency, results lag, sample pooling, disease prevalence, externally-acquired infections, symptom checking, and test cost on outcomes including case reduction and false positives.

### Findings

Increasing testing frequency was associated with a non-linear positive effect on cases averted over 100 days. While precise reductions in cumulative number of infections depended on community disease prevalence, testing every 3 days versus every 14 days (even with a lower sensitivity test) reduces the disease burden substantially. Pooling provided cost savings and made a high-frequency approach practical; one high-performing strategy, testing every 3 days, yielded per person per day costs as low as $1.32.

### Interpretation

A range of practically viable testing strategies emerged for schools and businesses. Key characteristics of these strategies include high frequency testing with a moderate or high

**Funding:** Authors [GL, NS, CK, DG, EB] are employees of UnitedHealth Group. Author [DG] also serves as the Chief of Infectious Disease for ProHealth NY, part of Optum. These funders provided support in the form of salaries for authors [GL, NS, CK, DG, EB], but did not have any additional role in the study design, data collection and analysis, decision to publish, or preparation of the manuscript. The specific roles of these authors are articulated in the 'author contributions' section.

**Competing interests:** Authors [GL, NS, CK, DG, EB] are employees of UnitedHealth Group; GL, NS, CK, and EB own stock in the company. DG is employed as the Senior Infectious Disease Fellow at UnitedHealth Group, Inc and serves as the Chief of Infectious Diseases for ProHealth NY an Optum Company. This does not alter our adherence to PLOS ONE policies on sharing data and materials.

sensitivity test and minimal results delay. Sample pooling allowed for operational efficiency and cost savings with minimal loss of model performance.

## Introduction

As schools and businesses re-open and attempt to stay open, promptly detecting people with infectious COVID-19 is essential, especially as the risk of transmission may be expected to increase as contact networks increase in size and complexity [1, 2]. Recommended actions to attenuate spread include symptom checking, monitoring underlying community prevalence, and responsive policy adjustment. In addition to robust public health measures, successful return to normalcy will be accelerated and hopefully sustained by optimal COVID-19 testing strategies. Despite being commonly recommended, little guidance suggests the right approach to testing and how best to balance cost, test selection, results delays, the value of sample pooling, and how changing local disease prevalence should inform strategy adjustments.

Throughout the pandemic the number and variety of tests for detecting active infection have steadily increased [3]. Current tests include nucleic acid amplification tests (NAATs) such as reverse-transcription or reverse transcription polymerase chain reaction (RT-PCR), template mediated amplification (TMA), nicking enzyme amplification reaction (NEAR), loop-mediated isothermal amplification (LAMP), nucleic acid hybridization, viral metagenomic sequencing, and CRISPR-based assays. Most Food and Drug Administration (FDA) Emergency Use Authorization (EUA) tests are approved for symptomatic patients, but not all are validated in an asymptomatic population [4]. Despite these scientific advancements, there is scant guidance on how to apply a specific technology in the context of the underlying population and the goal of testing, such as diagnosis of an individual versus surveillance of a group. Cost, turnaround time, accuracy, and convenience in sample collection all play a role in achieving a rate of testing that achieves a goal of detecting and preventing transmission in a cohort. A testing strategy is not feasible if the cost per test at the individual level is too high, or the time to obtain results is too long, resulting in possible transmission while positive test results are in transit or missing an opportunity to attend work or school if the result is negative. To increase test processing efficiency and reduce cost, pooling of samples is a potential solution provided there is minimal degradation in test performance due to dilution, but a strategy should be devised carefully. Successful pooling strategies rely on a clear understanding of the test's limit of detection (LOD), sensitivity, specificity, and the prevalence of disease in the population being tested [5].

Testing in large cohort settings such as schools and businesses that require continued surveillance can ensure that facilities remain open safely for the greatest number of people. We model various scenarios of test sensitivity and specificity, testing frequency, cost, and pooling to illustrate the range of practical and sustainable surveillance strategies.

## Methods

To compare the effects of test sensitivity and specificity, test frequency, and the impact of pooling we considered a classical epidemiological susceptible, infectious-asymptomatic, infectious-symptomatic, removed (SIR) compartmental model for the tested population. That is, individuals move from one compartment to another as they transition from susceptible to infectious to removed. To account for the introduction of infections from the surrounding community, we added a time-dependent term which represents the rate (in people/time) of infections from outside interactions continuously in time. With frequent testing, this external forcing drives the behavior of the model (Fig 1).

**Fig 1. Schematic representation of the model.** The model simulates testing for a common group of people who mix continuously in an institution (i.e., in a school or office) and are subject to the introduction of infection from the surrounding unmonitored community. The framework couples regular testing, described by a handful of tunable parameters, to a disease model. The disease model is dynamic in time, and infections may originate both from inside-the-institution mixing and from the surrounding community at varying rates depending on prevalence.

We examine two scenarios for this forcing. The first is a relatively low and more-or-less constant rate of introduced infections, with data from the 7-day rolling average of the case count in Fayette County, Pennsylvania for the 100 days beginning March 26, 2020, as reported in the *New York Times* [6]. This low-growth profile is reported as panel (a) in Figs 2–4. The second scenario used for high-growth external community prevalence is the seven-day rolling average of daily case counts in Miami-Dade County, Florida, for the 100 days beginning June 16, 2020. This profile is shown in panel (b) in Figs 2–4. In both profiles, we scaled the cases given by the relative population in our model, which we chose to be 1,500. It should be noted that the case counts in Miami-Dade County over this time period are outliers compared to case counts in other counties across the US over the past ten months. These cases are chosen for illustration to show the widest array of possible scenarios. To model symptom checking we solve the forced SIR model each day, and, at the end of the day, we remove the appropriate fraction of individuals from the infectious-symptomatic compartment. To model pooled testing with symptom tracking, when $\tau$ divides the day, in addition to removal due to symptom tracking from the infectious-symptomatic compartment, there is removal from both infectious compartments due to positive tests. The initial test is on day zero. To account for possible delays in receiving test results due to laboratory processing, we also allow for a delay parameter, $d$. When pooling samples, we adjust for test sensitivity and applied a linear deduction for pooling of 0.00323, consistent with minimal sample dilution or degradation in a nasal or nasopharyngeal sample [7]. Other deductions may be more appropriate in different settings, such as saliva sampling [8]. Our model allows for a varied percent of those that are infected to choose to comply with isolation protocols; in the scenarios presented, we set this tunable assumption to be perfect compliance. We assume the basic reproduction number $R_0$ is 2.5 and the average period of infectiousness is 4.5 days [9–12].

We do not include an "exposed" category as is often done for compartmental models but account for the shorter time a person is infectious rather than the longer period of time they are infected. Our model includes symptomatic and asymptomatic infectious individuals with daily symptom tracking. In the results that follow we assume 40% of infections are asymptomatic [13] and symptom tracking will catch 66% of symptomatic infections [14]. Individuals can

move directly from asymptomatic to removed or from asymptomatic, to symptomatic to removed.

The initial conditions are chosen from the average of population-scaled new confirmed cases reported by the *New York Times* for September 23, 2020 in a sample of counties scaled by average number of infectious days. This results in a starting value of 0.675 infections for a population of size 1,500. We take the conservative approach of assuming no one in the population has immunity to the virus based on previous infection. In the tests that follow we vary the testing frequency ($\tau$), delay in the return of results ($d$), number of samples pooled ($m$), sensitivity of the test on one sample, and specificity of the test. We computed the cost of each testing strategy at the per person per day level, over 100 days. When pooling ($m>1$), we assumed a simple 2-stage Dorfman testing process in which each individual in a positive pool is retested individually using a high-sensitivity diagnostic test at $100 per test. We then calculated the expected number of tests required to complete each round of testing. The complete scientific code is available as S1 File. All analysis was done using Julia v1.5.1 [15].

## Results

Fig 2 demonstrates scenarios of testing frequency at sensitivities of 98% with a two-day delay in receiving results during which mixing continues (Fig 2C and 2D), 98% sensitivity with no delay in receiving results (Fig 2E and 2F), and 60% sensitivity with no delay (Fig 2G and 2H) to simulate testing by various technologies such as PCR with lags between sample collection and centralized laboratory testing, antigen detection, and LAMP. As there are little data on the performance of some tests in asymptomatic people, we used more conservative sensitivity estimates aligning to published LOD for specific devices from the FDA [16]. The sawtooth pattern is the result of removal of infected persons from the population.

Any testing strategy is better than none at all, and as expected, tests with increased sensitivities perform better for a given time frequency. At the most lenient frequency considered, every 14 days, the number of infections is reduced approximately 21–56% (Table 1) compared to no testing at all. Each scenario can be explored comparatively. For example, at a test sensitivity of 80%, testing every day reduces the number of cumulative infections relative to no testing by 95.9–99.9% while testing every 14 days reduced the number of cumulative infections at day 100 relative to no testing by only 26.0–27.1%. Increased testing frequency results in a nonlinear decrease in cumulative infections over time, with daily testing resulting in the fewest cumulative infections at 100 days after implementing the testing strategy at any of the sensitivities shown. Importantly, at sensitivities of 98% our models predict that a two-day delay in results (by send-out PCR, for example) will result in just a 31% reduction in infections experienced at a 14-day testing frequency; however, as the testing frequency is increased, even with the two-day delay, the number of missed infections goes down rapidly to a 99% reduction from no testing at all at a daily testing frequency.

Next we looked at testing strategies that incorporate pooling. Fig 3 combines a weekly and every 3 day testing strategy with 98% sensitive tests with varying time delay, and pooling tests in samples of 2, 5, 10, and 30. Pooling potentially reduces the sensitivity of the tests, resulting in more missed infections. This can be overcome by an increase in test frequency, allowed by the cost savings of pooling. Fig 4 weighs cost against testing frequency and pooling size, both with confirmatory and without confirmatory testing of positive pools. Without confirmatory testing, the cost per person decreases dramatically.

## Discussion

Our findings demonstrate that it is not only critical to choose the right test in terms of performance in asymptomatic individuals, but to use the test in the defined population at the optimal

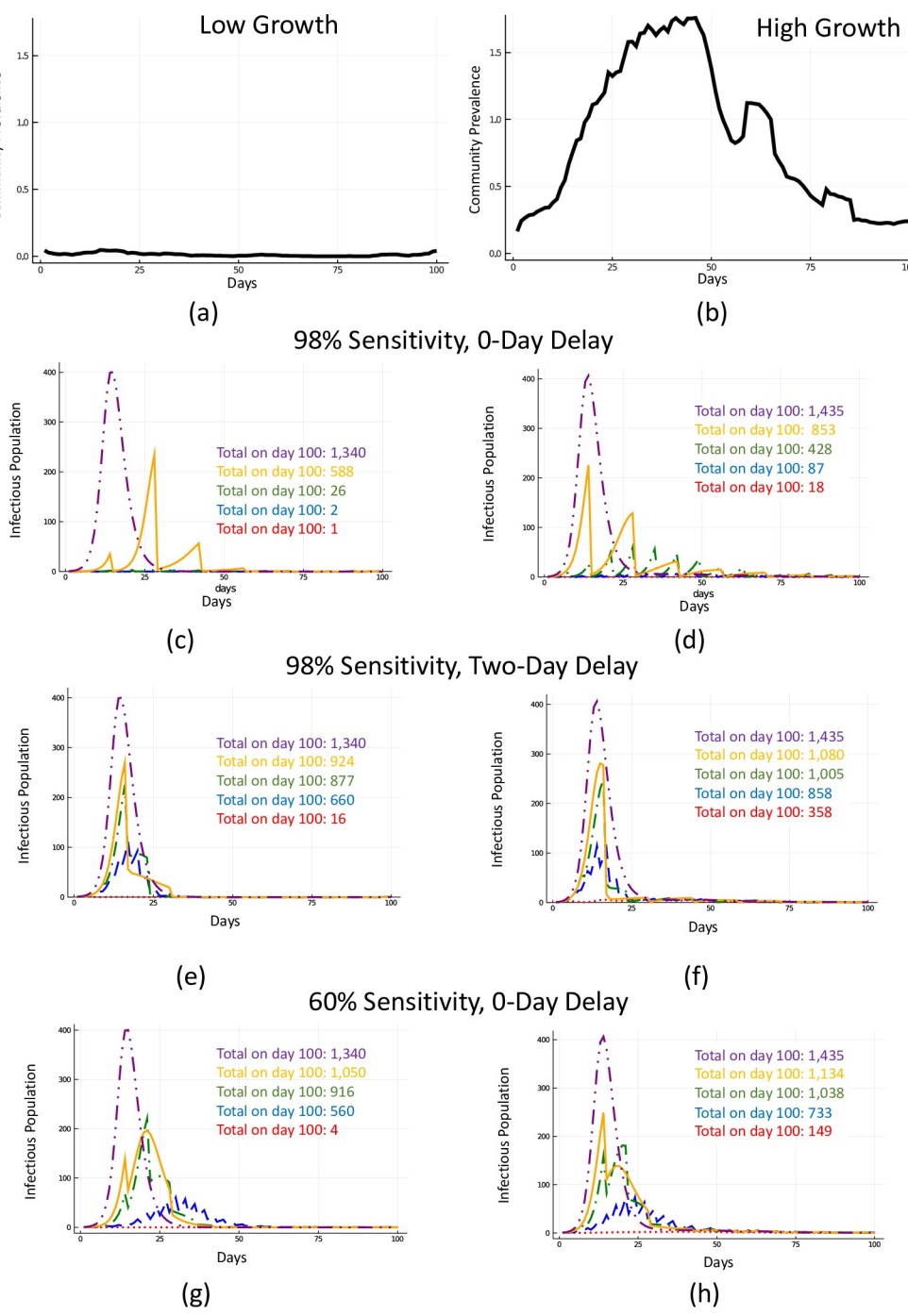

**Fig 2. Impact of testing frequency.** Two scenarios for community prevalence corresponding, relatively, to low and high rates of imported infections (Panels (a) and (b)). Testing with a test with 98% sensitivity with 0-day resulting delay amidst high and low community prevalence (Panels (c) and (d)). Testing with a test with 98% sensitivity with 2-day resulting delay amidst high and low community prevalence (Panels (e) and (f)). Testing with a test with 60% sensitivity with 0-day resulting delay amidst high and low community prevalence (Panels (g) and (h)). Purple (dash-dot-dot) corresponds to no testing, orange (solid) to testing every two weeks with daily symptom tracking, green for testing every week with daily symptom tracking (dash-dot), blue (dash) for testing every 3 days with daily symptom tracking, and red (dot) for daily testing and symptom tracking.

**Table 1. Selected testing strategies ranked by reduction in cumulative infections, with and without confirmatory testing, for scenarios costing less than $10 per person per day***.  Daily symptom tracking assumed.

| Sensitivity | Specificity | Delay (days) | Frequency (days) | Pool Size | Cumulative Infections Experienced (over 100 days) | Cumulative Infections Caught (over 100 days) | Cumulative False Positives (over 100 days) | Per Person, Per Day Cost Without Confirmatory Testing ($) | Per Person, Per Day Cost With Confirmatory Testing ($) | % Reduction in Cumulative Infections Experienced |
|---|---|---|---|---|---|---|---|---|---|---|
| **Community Prevalence: Low** | | | | | | | | | | |
| 0.98 | 0.995 | 0 | 3 | 5 | 2 | 5 | 255 | $ 7.92 | $ 8.78 | 99.8% |
| 0.98 | 0.995 | 0 | 3 | 10 | 3 | 5 | 255 | $ 3.96 | $ 5.65 | 99.8% |
| 0.98 | 0.995 | 0 | 3 | 30 | 5 | 8 | 255 | $ 1.32 | $ 6.20 | 99.7% |
| 0.98 | 0.995 | 0 | 7 | 2 | 35 | 68 | 112 | $ 8.40 | $ 8.63 | 97.4% |
| 0.98 | 0.995 | 0 | 7 | 5 | 100 | 180 | 112 | $ 3.36 | $ 4.21 | 92.6% |
| 0.98 | 0.995 | 0 | 7 | 10 | 241 | 388 | 111 | $ 1.68 | $ 4.23 | 82.0% |
| 0.98 | 0.995 | 0 | 7 | 30 | 515 | 628 | 110 | $ 0.56 | $ 6.58 | 61.5% |
| 0.6 | 0.9 | 0 | 3 | 1 | 560 | 536 | 5034 | $ 6.60 | $ 10.22 | 58.2% |
| 0.98 | 0.995 | 0 | 14 | 1 | 588 | 649 | 58 | $ 8.40 | $ 8.73 | 56.1% |
| 0.98 | 0.995 | 2 | 3 | 5 | 666 | 446 | 246 | $ 6.60 | $ 8.96 | 50.3% |
| 0.98 | 0.995 | 2 | 3 | 30 | 706 | 467 | 246 | $ 1.10 | $ 9.87 | 47.3% |
| 0.8 | 0.9 | 0 | 7 | 1 | 712 | 675 | 2195 | $ 7.00 | $ 8.76 | 46.9% |
| 0.98 | 0.995 | 2 | 7 | 2 | 877 | 542 | 104 | $ 7.00 | $ 7.66 | 34.5% |
| 0.98 | 0.995 | 2 | 7 | 5 | 880 | 542 | 104 | $ 2.80 | $ 4.18 | 34.3% |
| 0.98 | 0.995 | 2 | 7 | 10 | 884 | 543 | 104 | $ 1.40 | $ 3.61 | 34.0% |
| 0.98 | 0.995 | 2 | 7 | 30 | 899 | 545 | 104 | $ 0.47 | $ 4.52 | 32.9% |
| 0.6 | 0.9 | 0 | 7 | 1 | 916 | 638 | 2198 | $ 2.80 | $ 4.47 | 31.6% |
| 0.98 | 0.995 | 2 | 14 | 1 | 924 | 551 | 51 | $ 7.00 | $ 7.26 | 31.1% |
| 0.8 | 0.9 | 0 | 14 | 1 | 977 | 631 | 1171 | $ 3.50 | $ 4.44 | 27.1% |
| 0.6 | 0.9 | 0 | 14 | 1 | 1050 | 603 | 1171 | $ 1.40 | $ 2.30 | 21.6% |
| **Community Prevalence: High** | | | | | | | | | | |
| Sensitivity | Specificity | Delay (days) | Frequency (days) | Pool Size | Cumulative Infections Experienced (over 100 days) | Cumulative Infections Caught (over 100 days) | Cumulative False Positives (over 100 days) | Per Person, Per Day Cost Without Confirmatory Testing | Per Person, Per Day Cost With Confirmatory Testing | % Reduction in Cumulative Infections Experienced |
| 0.98 | 0.995 | 0 | 3 | 5 | 97 | 267 | 254 | $ 7.92 | $ 9.56 | 93.2% |
| 0.98 | 0.995 | 0 | 3 | 10 | 112 | 283 | 254 | $ 3.96 | $ 7.23 | 92.2% |
| 0.98 | 0.995 | 0 | 3 | 30 | 187 | 366 | 253 | $ 1.32 | $ 11.01 | 87.0% |
| 0.98 | 0.995 | 0 | 7 | 2 | 442 | 723 | 109 | $ 8.40 | $ 9.31 | 69.2% |
| 0.98 | 0.995 | 0 | 7 | 5 | 480 | 749 | 109 | $ 3.36 | $ 5.50 | 66.5% |
| 0.98 | 0.995 | 0 | 7 | 10 | 538 | 781 | 109 | $ 1.68 | $ 5.50 | 62.5% |
| 0.98 | 0.995 | 0 | 7 | 30 | 704 | 823 | 109 | $ 0.56 | $ 7.51 | 50.9% |
| 0.6 | 0.9 | 0 | 3 | 1 | 733 | 698 | 5015 | $ 6.60 | $ 10.28 | 48.9% |
| 0.98 | 0.995 | 0 | 14 | 1 | 853 | 845 | 57 | $ 8.40 | $ 8.79 | 40.5% |
| 0.8 | 0.9 | 0 | 7 | 1 | 859 | 799 | 2186 | $ 7.00 | $ 8.80 | 40.1% |
| 0.98 | 0.995 | 2 | 3 | 5 | 863 | 578 | 246 | $ 6.60 | $ 9.43 | 39.9% |
| 0.98 | 0.995 | 2 | 3 | 30 | 892 | 590 | 245 | $ 1.10 | $ 11.70 | 37.8% |
| 0.98 | 0.995 | 2 | 7 | 2 | 1006 | 624 | 103 | $ 7.00 | $ 7.72 | 29.9% |
| 0.98 | 0.995 | 2 | 7 | 5 | 1008 | 624 | 103 | $ 2.80 | $ 4.33 | 29.7% |
| 0.98 | 0.995 | 2 | 7 | 10 | 1013 | 625 | 103 | $ 1.40 | $ 3.90 | 29.4% |
| 0.98 | 0.995 | 2 | 7 | 30 | 1031 | 627 | 103 | $ 0.47 | $ 5.24 | 28.1% |
| 0.6 | 0.9 | 0 | 7 | 1 | 1038 | 720 | 2191 | $ 2.80 | $ 4.50 | 27.7% |

*(Continued)*

**Table 1.** (Continued)

| 0.8 | 0.9 | 0 | 14 | 1 | 1062 | 750 | 1158 | $ 3.50 | $ 4.50 | 26.0% |
|---|---|---|---|---|---|---|---|---|---|---|
| 0.98 | 0.995 | 2 | 14 | 1 | 1080 | 645 | 51 | $ 7.00 | $ 7.26 | 24.7% |
| 0.6 | 0.9 | 0 | 14 | 1 | 1134 | 681 | 1161 | $ 1.40 | $ 2.33 | 21.0% |

* Cost calculation assumes a test with a 98% sensitivity and 0-day delay in returning results costs $120, a 98% sensitive test with a 2-day delay in results costs $100, an 80% sensitive test costs $50, and a 60% sensitive test costs $20. All (true and false) positive tests are confirmed using a $100 test. The distribution of positive tests among pooled samples is uniform as is consistent with the homogeneous mixing assumptions of the SIR model, and we assume everyone in a pool that is positive will undergo a confirmatory test.

frequency to reduce the risk of case escalation. Optimization is further enhanced at the population level by understanding of underlying disease prevalence and utilization of pooling to reduce cost and increase efficiency. The "ideal" test strategy must be balanced with the practicalities of cost per person to ensure sustainability. For example, daily testing with a 60% sensitive test attenuates community spread, but at a cost of $30.11 per person per day with confirmatory testing, or $20.00 without, may not be possible. Using a 60% sensitive test less frequently reduces expense but sacrifices significant performance. A 98% sensitive test with no delay in results administered every 3 days with pools of 30 people, and no confirmatory test offered by the institution costs less than $1.50 per person per day, with high performance. Even with a highly specific (99.5%) test such as a PCR, in a low prevalence community with large pools, false positives may still become an issue. The previous example results in 253 false positives over 100 days, highlighting the importance of confirmatory testing. The model demonstrates that frequency of testing, test sensitivity, turn-around time, and the external community prevalence are all important factors to consider, and there is often more than one testing strategy to achieve the desired level of performance. The computational code is available as an on line supplement, and an easy-to-use web-based simulator to test various scenarios is available at https://calculator.unitedinresearch.com/. The ability to test different strategies under a variety of assumptions is especially important as we learn more about the performance of tests in asymptomatic populations over time.

With these scenarios in hand, institutions can make an informed operational choice, devise pods or cohorts to be tested by pooling and potentially isolated if positive, and create clear communication about a surveillance rationale. Acknowledging a dynamic community prevalence, the model can be re-run, and the testing strategy can be optimized to maximize benefit at the lost cost and least amount of disruption.

The frequency of test usage to minimize amplification of infection and allow schools and worksites to remain open is an important factor. Given the cost of high frequency testing, we demonstrate the value of pooling of samples to increase efficiency, particularly in areas with lower population prevalence. As background prevalence increases, the value of pooling diminishes as the likelihood of a positive pool will rise, but even a pool of two to three samples results in a dramatic reduction in the need for individual sample analysis. As noted above, with an extremely low prevalence, even in the case of a 99.5% specific test, false positives are much more likely than true positives and confirmatory testing may be necessary. A 90% specificity test would result in an untenable number of false positives over the course of 100 days without confirmatory testing. As shown in Fig 4, in order to achieve a minimal cost approach that includes confirmatory testing, one must balance pool size with frequency. Without confirmatory testing, costs drop dramatically (Fig 4G and 4H). The Dorfman protocol for pooled testing we use is suboptimal compared to a sequential split-pool strategy [17]. More sophisticated confirmatory testing strategies exist that would further lower costs and reduce the likelihood

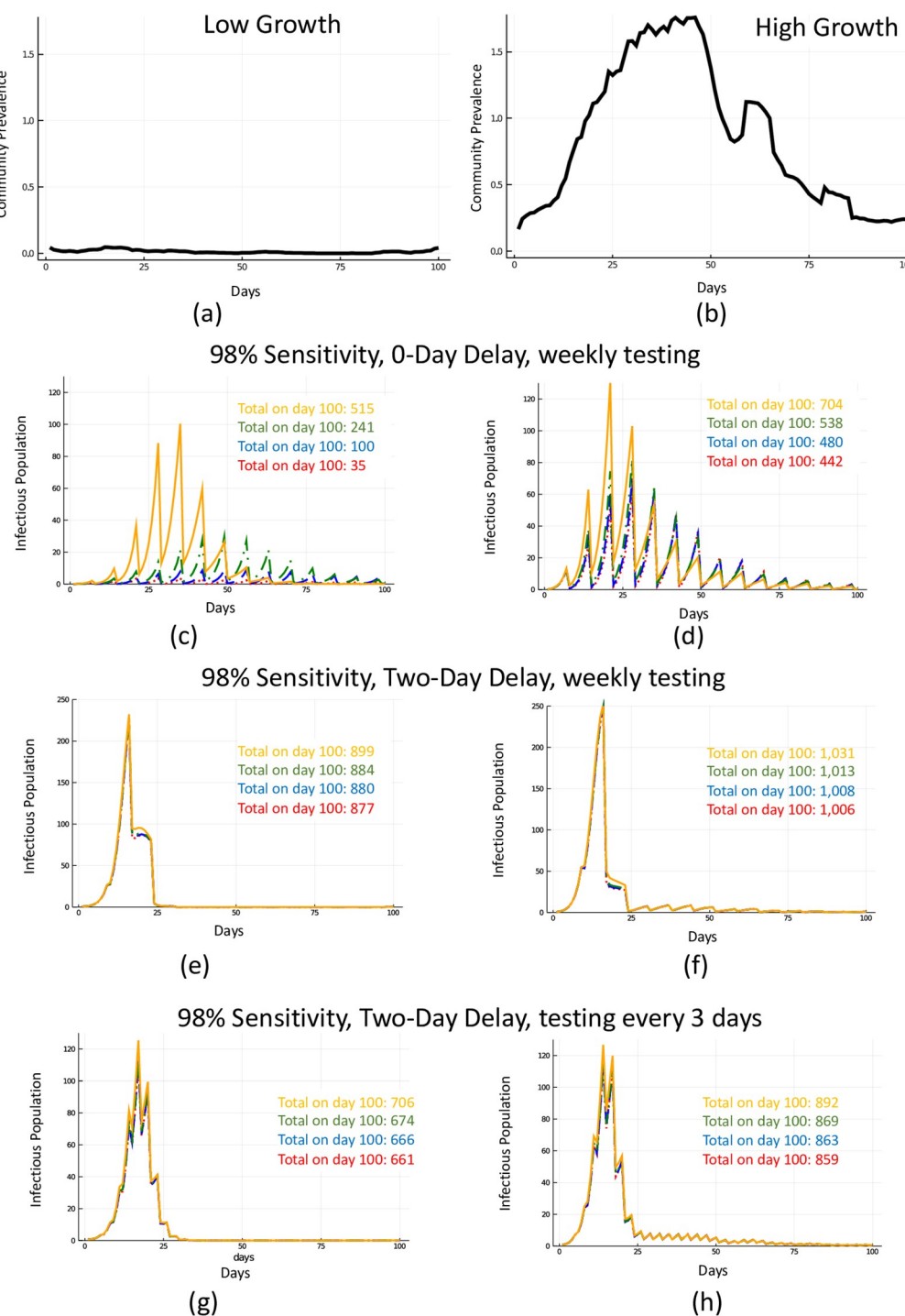

**Fig 3. Effect of pool size.** Two scenarios for community prevalence corresponding, relatively, to low and high rates of imported infections (Panels (a) and (b)). Testing weekly with a test with 98% sensitivity with 0-day resulting delay with daily symptom tracking amidst high and low community prevalence (Panels (c) and (d)). Testing weekly with a test with 98% sensitivity with 2-day resulting delay with daily symptom tracking amidst high and low community prevalence (Panels (e) and (f)). Testing every 3 days with a test with 98% sensitivity with 2-day resulting delay with daily symptom tracking amidst high and low community prevalence (Panels (g) and (h)). Orange lines (solid) correspond to 30 samples pooled, green (dash-dot) to ten samples pooled, blue (dash) to five samples pooled, and red (dot) to 2 samples pooled.

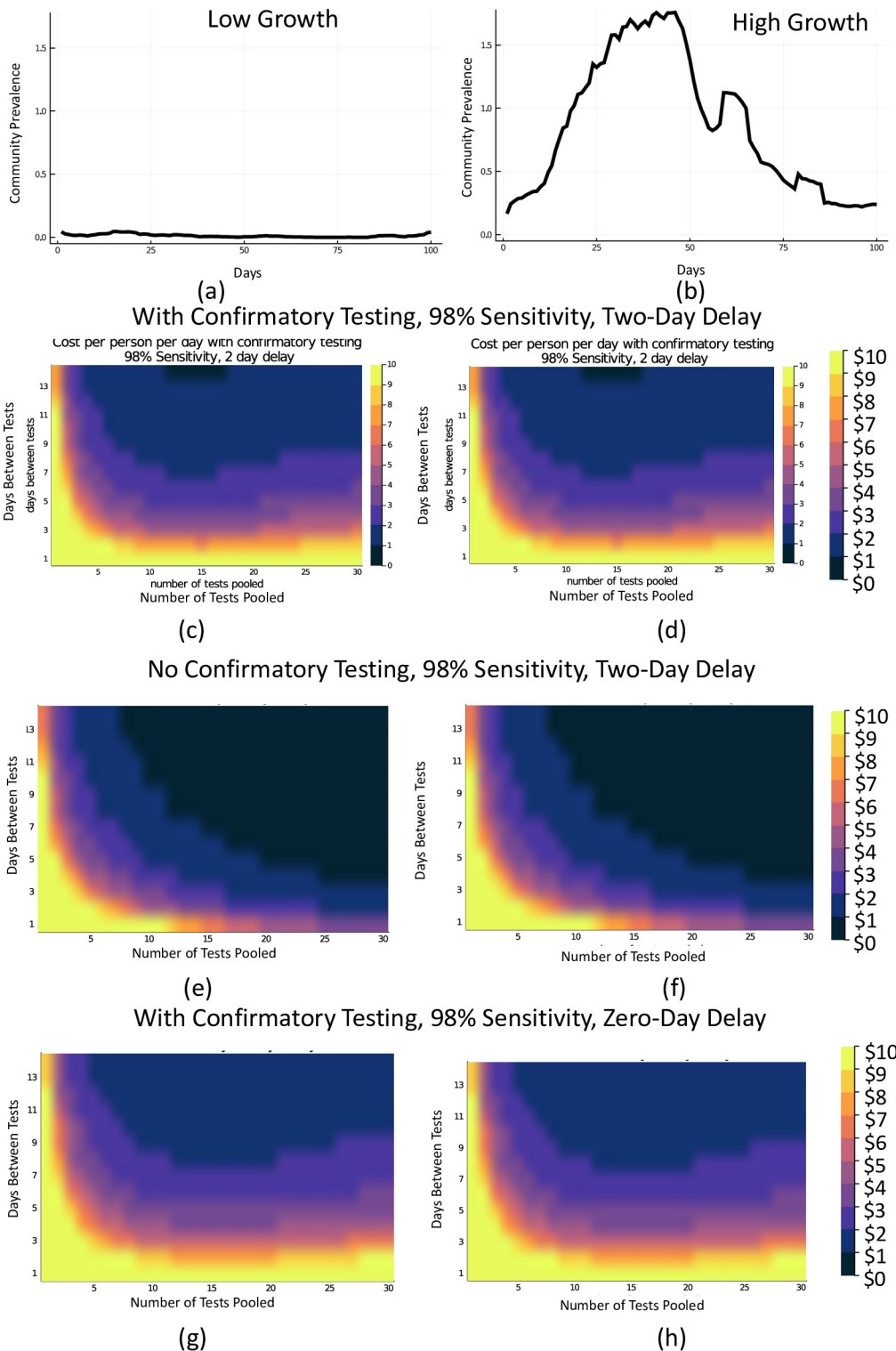

**Fig 4. Cost comparison map for various pooling and frequency scenarios with and without confirmatory testing.**
Use case of a test with 98% sensitivity and 99.5% specificity with a 2-day result delay costing $100 and a 98% sensitive test with 99.5% specificity and a 0 day result delay costing $120 with free daily symptom tracking. In (c, d, g, h) every person in a positive pool is retested for confirmation and in (e, f) no confirmatory testing is done. We assume all confirmatory tests cost $100. Colors correspond to cost per person per day in dollars.

that uninfected individuals are sent home, such as sub-pooling of positive pools without individual level testing, each with benefits and disadvantages [18, 19].

This study confirms and extends previous work. Paltiel et al. [20] considered a compartment-based model simulating an abbreviated 80-day semester in a highly-residential college-campus-type setting. Across all scenarios considered, test frequency was more associated with cumulative infection than test sensitivity. That modeling exercise also suggested that symptom-based screening alone is insufficient to contain an outbreak under any of the scenarios considered. Using a model for viral loads in individuals, Larremore et al. [21] studied surveillance effectiveness using an agent-based modeling framework which accounts for test sensitivities, frequency, and sample-to-answer reporting time. The results indicate that frequency of testing and the speed of reporting are the principal contributors to surveillance effectiveness. The results also suggest that the impact of high sensitivity on surveillance effectiveness is relatively small.

Populations housed in long-term care facilities are especially vulnerable to COVID-19; surveillance programs designed for these settings may have different goals and tolerances for infection risk than those designed to maintain functionality for other institutions. Smith and colleagues [7] built a complex modeling framework for long-term care facilities including simulations of the detailed inter-individual contact networks describing patient-staff interactions in such settings. This work showed that symptom-based screening by itself had limited effectiveness. Testing upon admission detected most asymptomatic cases upon entry but missed potential introductions from staff. Random daily testing was determined to be, overall, an inefficient use of resources. This points to the opportunity for pooled testing as an effective and efficient COVID-19 surveillance strategy for long-term care facilities with limited resources.

Since our work focuses on screening and not performing diagnostic testing, the actual sensitivity of the various available COVID tests for this purpose is not entirely clear. The original testing approaches for COVID-19 focused on the high sensitivity required for diagnosis by clinicians in all stages of the acute period of COVID-19 through detection of SARS-CoV-2 RNA performed on patients with a high pretest probability of disease. This paradigm focused on high sensitivity tests with the performance feature of very low NAAT detectable units/mL (NDU/mL) with a goal of diagnosing patients even if past the contagious period. These tests were not optimized nor validated in terms of sensitivity for the detection of infectious individuals that might spread disease in schools, the workplace or other social situations.

Several studies looking at the ability to culture virus from samples collected from infected individuals have established that RNA copy numbers of 1,000,000 RNA copies/ml or higher are required for any consistent success in viral culture [22–26]. Based on contact tracing, this defined window of elevated RNA copy numbers starting 2–3 days prior to onset of symptoms and ending 5–9 days after symptom onset corresponds to most if not all cases of transmission. Studies of asymptomatic spreading suggests a very similar window of transmissibility during this period of time when RNA copy numbers are 1,000,000 copies/ml or higher [12, 27, 28]. Given that RT-PCR testing can have a sensitivity or LOD as low as <1,000 RNA copies/mL (1,000 NDU), there should ample performance in testing technology to leverage high-volume, high-frequency pooling, provided samples are not diluted by storage or buffering media beyond the minimum LOD when employed to detected asymptomatic but infectious individuals [29].

Our work has a number of limitations. The SIR compartmental model provides a simplified representation of the natural history of the disease. For example, it assumes uniform mixing of the population being tested and a uniform distribution of likelihood of a positive test. The model is formulated at a population level; it does not permit the tracking of individuals. For example, we cannot incorporate the change in test sensitivity with time from infection [30]. In

a low population prevalence, we expect a high number of false positives given assumed speci-ficities of 99.5% and 90%. Individuals who recover from the disease are granted permanent immunity in our model, although the risk of reinfection now appears possible [31–36]. Our pooling model assumed nasal or naso-pharyngeal swab samples. Because of the nature of saliva, the small sensitivity deduction assumption in our model may not be valid due to greater sample dilution [37]. Finally, the model does not naturally incorporate phased, pulsed, or par-tial testing (1st graders on Monday, 2nd graders on Tuesday, etc.). To account for this we sug-gest users model the smaller groups and multiply results rather than attempt to run scenarios on the full population.

Despite these limitations, sensitivity, pooling, and frequency modeling can guide institu-tions on best-fit testing strategies that align to their practical constraints. Organizations can apply this model to determine their best testing strategy given current community prevalence and operational and financial resources that enable sustained testing to stay safely open during the pandemic.

## Supporting information

**S1 File. Technical details about the model.**
(PDF)

## Author Contributions

**Conceptualization:** Gregory D. Lyng, Natalie E. Sheils, Daniel O. Griffin, Ethan M. Berke.

**Data curation:** Gregory D. Lyng, Natalie E. Sheils.

**Formal analysis:** Gregory D. Lyng, Natalie E. Sheils, Caleb J. Kennedy, Ethan M. Berke.

**Investigation:** Gregory D. Lyng, Natalie E. Sheils, Caleb J. Kennedy, Daniel O. Griffin, Ethan M. Berke.

**Methodology:** Gregory D. Lyng, Natalie E. Sheils, Caleb J. Kennedy, Daniel O. Griffin, Ethan M. Berke.

**Supervision:** Ethan M. Berke.

**Visualization:** Natalie E. Sheils.

**Writing – original draft:** Gregory D. Lyng, Natalie E. Sheils, Caleb J. Kennedy, Daniel O. Grif-fin, Ethan M. Berke.

**Writing – review & editing:** Gregory D. Lyng, Natalie E. Sheils, Caleb J. Kennedy, Daniel O. Griffin, Ethan M. Berke.

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
