## [Decision Letter · Decision Letter 0]

22 Feb 2021

PONE-D-20-36267

Identifying Optimal COVID-19 Testing Strategies for Schools and Businesses: Balancing Testing Frequency, Individual Test Technology, and Cost

PLOS ONE

Dear Dr. Berke,

Thank you for submitting your manuscript to PLOS ONE. After careful consideration, we feel that it has merit but does not fully meet PLOS ONE’s publication criteria as it currently stands. Therefore, we invite you to submit a revised version of the manuscript that addresses the points raised during the review process.

I apologize for the delay in making a decision on your manuscript. This was all due to problems in securing the required minimum number of reviewers. 

Please attend to all the comments and and observations made by the reviewer in addition to these I make below:

1. Ensure that your referencing style conforms to that prescribed in the PLOS One guidelines.

2. Provide references for the tests you mention in lines 53 to 56.

3. Figure 1 title should read " Schematic representation of the model"

4. Tables 2, 3 and 4 are presenting results. Should it be not appropriate that they are presented in the results section instead of results?

5. Include the word "sensitivity" after the percentages in lines 62 and 63.

We look forward to receiving your revised manuscript.

Kind regards,

Martin Chtolongo Simuunza, PhD

Academic Editor

PLOS ONE

Additional Editor Comments:

Journal Requirements:

We note that one or more of the authors are employed by a commercial company: UnitedHealth Group Inc, ProHealth Care.

2.1. Please provide an amended Funding Statement declaring this commercial affiliation, as well as a statement regarding the Role of Funders in your study. If the funding organization did not play a role in the study design, data collection and analysis, decision to publish, or preparation of the manuscript and only provided financial support in the form of authors' salaries and/or research materials, please review your statements relating to the author contributions, and ensure you have specifically and accurately indicated the role(s) that these authors had in your study. You can update author roles in the Author Contributions section of the online submission form.

2.2. Please also provide an updated Competing Interests Statement declaring this commercial affiliation along with any other relevant declarations relating to employment, consultancy, patents, products in development, or marketed products, etc.  

Reviewers' comments:

Reviewer's Responses to Questions

**Comments to the Author**

1. Is the manuscript technically sound, and do the data support the conclusions?

Reviewer #1: Yes

2. Has the statistical analysis been performed appropriately and rigorously? 

Reviewer #1: Yes

3. Have the authors made all data underlying the findings in their manuscript fully available?

Reviewer #1: Yes

4. Is the manuscript presented in an intelligible fashion and written in standard English?

Reviewer #1: Yes

5. Review Comments to the Author

Reviewer #1: This is very good, original, and profound research work done and a well written manuscript paper. My review is therefore limited to the following minor comments only.

In the supplementary material detailing the model used, under testing strategies; for effect of Symptom tracking alone, you mention that you assume 78% of cases entering a symptomatic phase are caught and a fraction b of these cases self isolate. However, in the manuscript in line 144 – 145 you mention that for the results obtained you had assumed that symptom tracking will catch 66% of symptomatic infections. Kindly clarify this disparity in percentages considered. Wouldn’t it give a substantial disparity in the simulation results obtained as well?

For a population as small as the one modelled in this study (1500 subjects), a stochastic model may be more appropriate. Was this considered?

As alluded to in the manuscript, asymptomatic individuals may also reach just as high detectable viral load levels for the various testing technologies. However, when it comes to infectiousness, some corner of available literature on COVI-19 and earlier communications from the United Nations suggest that the asymptomatic individuals may not be as infectious as symptomatic individuals hence their per capita rate of effective contacts I.e (β) may not be the same as for individuals with symptomatic infection. This has not been considered in the model shown in the supplementary material in 3.2 (Equations). It may not be necessary to consider this at this point but chances are that incorporating this may produce different results and this scenario may be a more accurate representation of the natural history of the infection.

The “in text” referencing should be written in square brackets e.g as [14] and not as superscripts e.g as 14, 23.

The sentence between lines 42 and 44 reads’ “As schools and businesses re-open and attempt to stay open, promptly detecting people with infectious COVID-19 is essential, especially as the risk of transmission is expected to increase with colder weather, more time indoors, and closer contact with others….”.

My impression was that as economies open, people interact more outdoors as they are more frequently away from the indoor safety of their homes such as during lock downs. Perhaps just check that statement again on the part that says “more time indoors”.

The sentence between lines 77 and 79 should read as “To compare the effects of test sensitivity and specificity, test frequency, the impact of pooling, and other key factors influencing testing strategy, we considered a classical epidemiological susceptible, infectious-asymptomatic, infectious-symptomatic, removed (SIR) compartmental model for the tested population.

The sentence describing beta (β) is incomplete in the supplementary material detailing the model. It only partly says that beta is the contact rate. I believe beta can be more accurately described as the per capita rate of effective contacts.

6. PLOS authors have the option to publish the peer review history of their article (what does this mean?). If published, this will include your full peer review and any attached files.

Reviewer #1: No

---

## [Author Response · Author response to Decision Letter 0]

25 Feb 2021

February 24, 2021

Martin Chtolongo Simuunza, PhD

Academic Editor

PLOS ONE

Dear Dr. Simuunza:

 We appreciate the comments and suggestions of the reviewer and have incorporated their recommendations or provided alternative considerations. Below is our response to their comments in a point-by-point format, with the original reviewer comment in italics.

EDITOR COMMENTS:

 Ensure that your referencing style conforms to that prescribed in the PLOS One guidelines.

References have been updated to match PLOS One formatting guidelines. 

 Provide references for the tests you mention in lines 53 to 56.

Reference added: 

A. La Marca, M. Capuzzo, T. Paglia, L. Roli, T. Trenti, and S. M. Nelson, “Testing for SARS-CoV-2 (COVID-19): a systematic review and clinical guide to molecular and serological in-vitro diagnostic assays,” Reproductive BioMedicine Online, vol. 41, no. 3, pp. 483–499, Sep. 2020, doi: 10.1016/j.rbmo.2020.06.001.

 Figure 1 title should read " Schematic representation of the model"

Updated. 

 Tables 2, 3 and 4 are presenting results. Should it be not appropriate that they are presented in the results section instead of results?

There is only one table in the manuscript. We assume this refers to the figures. We have adjusted the placement of the figures so that they are in the Results section. 

 Include the word "sensitivity" after the percentages in lines 62 and 63.

We added sensitivity in lines 234 and 235, which is what we believe you were referring to.

 Thank you for stating the following in the Competing Interests section:

We note that one or more of the authors are employed by a commercial company: UnitedHealth Group Inc, ProHealth Care.

2.1. Please provide an amended Funding Statement declaring this commercial affiliation, as well as a statement regarding the Role of Funders in your study. If the funding organization did not play a role in the study design, data collection and analysis, decision to publish, or preparation of the manuscript and only provided financial support in the form of authors' salaries and/or research materials, please review your statements relating to the author contributions, and ensure you have specifically and accurately indicated the role(s) that these authors had in your study. You can update author roles in the Author Contributions section of the online submission form. 

“The funder provided support in the form of salaries for authors [insert relevant initials], but did not have any additional role in the study design, data collection and analysis, decision to publish, or preparation of the manuscript. The specific roles of these authors are articulated in the ‘author contributions’ section.

We have updated the funding statement to read: 

Authors [GL, NS, CK, DG, EB] are employees of Optum Labs at UnitedHealth Group. Author [DG] also serves as the Chief of Infectious Disease for ProHealth NY, part of Optum. These funders provided support in the form of salaries for authors [GL, NS, CK, DG, EB], but did not have any additional role in the study design, data collection and analysis, decision to publish, or preparation of the manuscript. 

Please also provide an updated Competing Interests Statement declaring this commercial affiliation along with any other relevant declarations relating to employment, consultancy, patents, products in development, or marketed products, etc.

GL is an employee of UnitedHealth Group and owns stock in the company. DG is employed as the Senior Infectious Disease Fellow at the commercial company, UnitedHealth Group, Inc and serves as the Chief of Infectious Diseases for ProHealth NY an Optum Company.

 Please include captions for your Supporting Information files at the end of your manuscript, and update any in-text citations to match accordingly. Please see our Supporting Information guidelines for more information: http://journals.plos.org/plosone/s/supporting-information.

We have added a caption for the supplementary file. 

REVIEWER COMMENTS

Reviewer #1: This is very good, original, and profound research work done and a well written manuscript paper. My review is therefore limited to the following minor comments only.

 In the supplementary material detailing the model used, under testing strategies; for effect of Symptom tracking alone, you mention that you assume 78% of cases entering a symptomatic phase are caught and a fraction b of these cases self isolate. However, in the manuscript in line 144 – 145 you mention that for the results obtained you had assumed that symptom tracking will catch 66% of symptomatic infections. Kindly clarify this disparity in percentages considered. Wouldn’t it give a substantial disparity in the simulation results obtained as well?

The supplemental information has been updated to reflect the assumption that symptom tracking will catch 66% of symptomatic infections. The source of this choice is the Nature Medicine article below: 

C. Menni et al., “Real-time tracking of self-reported symptoms to predict potential COVID-19,” Nature Medicine, vol. 26, no. 7, Art. no. 7, Jul. 2020, doi: 10.1038/s41591-020-0916-2.

However, as the reviewer notes, this value certainly has an impact on the simulation results, and there is continuing discussion in the literature about the efficacy of symptom tracking. For this reason, this is an adjustable parameter in the online calculator. 

 For a population as small as the one modelled in this study (1500 subjects), a stochastic model may be more appropriate. Was this considered?

Our aim in this project was to build the simplest model we could think of that could incorporate pooling and deal with the realities of a dynamic underlying (and unknown!) community prevalence of disease. For this reason, we built the model on basic SIR-type deterministic dynamics. One of the reasons for this was to ensure that the model (and its limitations) could easily be explained to policy makers and others (school superintendents, long-term care facility administrators, parents,…). For this reason, we did not consider variations or other modeling approaches, e.g., stochastic models or agent-based approaches, that might better capture some elements of the situation. 

 As alluded to in the manuscript, asymptomatic individuals may also reach just as high detectable viral load levels for the various testing technologies. However, when it comes to infectiousness, some corner of available literature on COVI-19 and earlier communications from the United Nations suggest that the asymptomatic individuals may not be as infectious as symptomatic individuals hence their per capita rate of effective contacts I.e (β) may not be the same as for individuals with symptomatic infection. This has not been considered in the model shown in the supplementary material in 3.2 (Equations). It may not be necessary to consider this at this point but chances are that incorporating this may produce different results and this scenario may be a more accurate representation of the natural history of the infection.

It is true that the model does not incorporate a distinction in infectiousness between symptomatic and asymptomatic individuals. In the name of simplicity, we assume, however, asymptomatic infectious individuals are as contagious as symptomatic ones. Notably, this “stacks the deck” against any testing program one might design. Thus, this is a conservative simplifying assumption. While we certainly don’t advocate being wasteful with testing capacity, our objective was to err on the side of overbuilding testing programs. 

 The “in text” referencing should be written in square brackets e.g as [14] and not as superscripts e.g as 14, 23.

References have been updated to match PLOS One formatting guidelines. 

 The sentence between lines 42 and 44 reads’ “As schools and businesses re-open and attempt to stay open, promptly detecting people with infectious COVID-19 is essential, especially as the risk of transmission is expected to increase with colder weather, more time indoors, and closer contact with others….”.

My impression was that as economies open, people interact more outdoors as they are more frequently away from the indoor safety of their homes such as during lock downs. Perhaps just check that statement again on the part that says “more time indoors”.

When this draft was originally written, during fall in the northern hemisphere, there was great concern about colder temperatures in winter leading to more indoor activities (and potentially more infections). For example, in some localities in the US, some restaurants stayed open in the fall by utilizing outdoor dining. The reviewer is correct, of course, that re-opening schools and businesses will also lead to the more contacts between individuals from distinct households. 

The sentence has been rewritten to read: “As schools and businesses re-open and attempt to stay open, promptly detecting people with infectious COVID-19 is essential, especially as the risk of transmission may be expected to increase as contact networks increase in size and complexity.”

 The sentence between lines 77 and 79 should read as “To compare the effects of test sensitivity and specificity, test frequency, the impact of pooling, and other key factors influencing testing strategy, we considered a classical epidemiological susceptible, infectious-asymptomatic, infectious-symptomatic, removed (SIR) compartmental model for the tested population.

The sentence describing beta (β) is incomplete in the supplementary material detailing the model. It only partly says that beta is the contact rate. I believe beta can be more accurately described as the per capita rate of effective contacts.

The description of β in the supplementary material has been updated.

---

## [Decision Letter · Decision Letter 1]

5 Mar 2021

Identifying Optimal COVID-19 Testing Strategies for Schools and Businesses: Balancing Testing Frequency, Individual Test Technology, and Cost

PONE-D-20-36267R1

Dear Dr. Berke,

We’re pleased to inform you that your manuscript has been judged scientifically suitable for publication and will be formally accepted for publication once it meets all outstanding technical requirements.

Kind regards,

Martin Chtolongo Simuunza, PhD

Academic Editor

PLOS ONE

Additional Editor Comments (optional):

Reviewers' comments:

Reviewer's Responses to Questions

**Comments to the Author**

1. If the authors have adequately addressed your comments raised in a previous round of review and you feel that this manuscript is now acceptable for publication, you may indicate that here to bypass the “Comments to the Author” section, enter your conflict of interest statement in the “Confidential to Editor” section, and submit your "Accept" recommendation.

Reviewer #1: All comments have been addressed

2. Is the manuscript technically sound, and do the data support the conclusions?

Reviewer #1: Yes

3. Has the statistical analysis been performed appropriately and rigorously? 

Reviewer #1: Yes

4. Have the authors made all data underlying the findings in their manuscript fully available?

Reviewer #1: Yes

5. Is the manuscript presented in an intelligible fashion and written in standard English?

Reviewer #1: Yes

6. Review Comments to the Author

Reviewer #1: Major comment:

All my corrections have been sufficiently attended to. The paper is now well written and the methodology scientifically sound.

Minor corrections:

- Line 283 – 287 recheck for correct grammar.

7. PLOS authors have the option to publish the peer review history of their article (what does this mean?). If published, this will include your full peer review and any attached files.

Reviewer #1: No

---

## [Editor Report · Acceptance letter]

9 Mar 2021

PONE-D-20-36267R1 

Identifying optimal COVID-19 testing strategies for schools and businesses: Balancing testing frequency, individual test technology, and cost 

Dear Dr. Berke:

I'm pleased to inform you that your manuscript has been deemed suitable for publication in PLOS ONE. Congratulations! Your manuscript is now with our production department. 

Kind regards, 

on behalf of

Dr. Martin Chtolongo Simuunza 

Academic Editor

PLOS ONE